# *Peronema canescens* as a Source of Immunomodulatory Agents: A New Opportunity and Perspective

**DOI:** 10.3390/biology13090744

**Published:** 2024-09-22

**Authors:** Ahmad Hafidul Ahkam, Yasmiwar Susilawati, Sri Adi Sumiwi

**Affiliations:** 1Department of Pharmacology and Clinical Pharmacy, Faculty of Pharmacy, Padjadjaran University, Sumedang 45363, Indonesia; ahmad23011@mail.unpad.ac.id; 2Department of Biology Pharmacy, Faculty of Pharmacy, Padjadjaran University, Sumedang 45363, Indonesia; yasmiwar@unpad.ac.id; 3The Herbal Studies Center, Faculty of Pharmacy, Padjadjaran University, Sumedang 45363, Indonesia

**Keywords:** immunomodulatory agent, lamiaceae, peronemin, TOGA

## Abstract

**Simple Summary:**

This review describes the potential of *Peronema canescens* extract as a source of immunomodulatory agents. This plant extract also has a wide range of bioactivities; however, information and research on this matter are very limited. This review discusses the bioactivity of *Peronema canescens* extract, its potential as a source of immunomodulatory agents, and suggestions for its development.

**Abstract:**

Immunomodulators are pivotal in managing various health conditions by regulating the immune response by either enhancing or suppressing it to maintain homeostasis. The growing interest in natural sources of immunomodulatory agents has spurred the investigation of numerous medicinal plants, including *Peronema canescens*, commonly known in Asia as sungkai. Traditionally used for its medicinal properties in Southeast Asia, *Peronema canescens* belongs to the Verbenaceae family and has garnered significant attention. This review discusses the immunomodulatory activity of the active compounds in *Peronema canescens* and explores the potential directions for future research.

## 1. Introduction

Traditional medicine is an alternative treatment method still used today that has a high economic value. One of the traditional medicinal plants with a high bioactivity but that is still limited in traditional use or development of its active compounds is *Peronema canescens*. The therapeutic features of *Peronema canescens* may be due to its rich phytochemical composition including alkaloids, flavonoids, saponins, and tannin steroids, which exert antioxidant, antibacterial, and anticancer effects. Interestingly, newly identified compounds like peronemin have been isolated from this plant, and each one plays a vital role in its pharmacological profile [1,2].

One of the current growing treatment trends is using the immune system to fight pathogens. Drugs that target the bioactivity of the immune system are known as immunomodulatory agents. Immunomodulators play key roles in treating various diseases by manipulating the immune response by either stimulating or suppressing it to achieve homeostasis. The growing interest in natural sources of immunomodulatory activities has resulted in the study of several medicinal plants, among which, *Peronema canescens* is at the forefront; it belongs to the family Verbenaceae and has long been utilised for its medicinal value in various cultures, specifically in Southeast Asia [3].

While *Peronema canescens* has long been used traditionally [3], scientific studies on it are relatively recent. Modern pharmacological research has tended to confirm its traditional applications and expose it, inter alia, as a potential immunomodulator with large therapeutic promise. The present review is an attempt at a detailed examination of the botanical characteristics, phytochemical constituents, and pharmacological properties of *Peronema canescens*, emphasising its potential for becoming a source of immunomodulatory agents. This review describes the botanical profile and bioactivity of *Peronema canescens* extract related to its potential as an immunomodulator, as well as additional information on preserving and how to develop extracts from this plant to underline its importance in current pharmacology and showcase future promises of therapeutic applications.

## 2. Botanical Profile of *Peronema canescens* Jack

*Peronema canescens*, Bahasa Indonesia (sungkai), of the family Verbenaceae, has recently attracted much attention because of its phytochemical composition and potential pharmacological activities [4].

The Verbenaceae family is a diverse and key botanical entity, with around 2600 species arranged into 100 genera and a pantropical distribution [5]. The synthesis of ursolic acid and oleanolic acid, typical for various families of plants, including Verbenaceae, underscores its important biochemical profile [6]. Renowned for its constituents with important bioactive properties, the Verbenaceae flourish in pure and open forests [7,8], hosting rust fungi from different families unrelated to the genus Physopella [9]. Verbenaceae, a plant family with a rich history in Ayurvedic and indigenous medicinal systems, has a total of approximately 200 species of herbs, shrubs, and small trees, thereby being a cornerstone of traditional medicine [10,11]. More importantly, Verbenaceae is typified by variable habitat, significant bioactivity, and major components such as volatile monoterpenoids, sesquiterpenoids, luteolin, thymol, carvacrol, phenols, and 4-carboxylated iridoids [12,13,14,15,16]. The Verbenaceae family is widespread with major representations in the tropical and subtropical regions, also extending into temperate zones in Europe, South America, Africa, Southeastern Asia, Australia, and Micronesia [17].

Some studies and databases place the genus Peronema in the Lamiaceae rather than in the Verbenaceae family [18,19]. Lamiaceae is an extended family of plants representing a large degree of biological and medicinal activities; thus, it is considered an important family in herbal medicine [20]. Several species from this family are classified as aromatic spices such as *Thymus* sp. with its essential oil containing thymol, geraniol, and linalool [21]; *Oreganum* sp., the Origano symbiont, with its origanum oil containing thymol, terpinen-4-ol, and cymene [22]; *Mentha* sp. (mint) with its essential oil containing menthol, menthone, and isomenthone [23]; and savoury basil, sage, rosemary, and lemon balm [24].

The botanical and phytochemical profile of *Peronema canescens* depicts a plant with a rich composition that is of potential significance to pharmacology. This has generally been better placed in perspective by its membership in the bioactive and diverse family of Verbenaceae. With its wide global distribution and broad representation in traditional medicine, this places *Peronema canescens* and its related species in good perspective. The fact that some sources place it under the Lamiaceae family only underlines the intricate nature of its botanical taxonomy. The interest in *Peronema canescens* is enhanced by the prominence of Lamiaceae in herbal medicine, blended with an array of biologically active compounds within this family. The more one understands the intricacies of the phytochemical makeup of this plant, the more its potential applications seem to be promising in pharmacology and medicine, therefore warranting further research and investigation.

## 3. Phytochemical and Pharmacological Studies of *Peronema canescens* Jack

*Peronema canescens* is a member of the Verbenaceae family, which is reported to have great potential in the medical field [25]. Ethnobotanically, the *Peronema canescens* plant has been used as a traditional herbal plant in communities in Kalimantan, especially to treat malaria [26]. It is also used to improve the immune system, treat fever, and help with postpartum recovery [3].

Various analytical methods have been employed to identify and quantify the phytochemical constituents of *Peronema canescens* such as Fourier Transform Infrared (FTIR) analysis, chromatography, and qualitative phytochemical screening [27]. Additionally, in silico studies have predicted the bioactive compounds and potential immunomodulatory effects of *Peronema canescens*, providing valuable insights into the molecular interactions and mechanisms of action of its bioactive constituents [28].

*Peronema canescens* is characterised by the presence of alkaloids, flavonoids, saponins, tannins, and steroids in its leaves [25,29]. More advanced studies on the phytochemical constituents of *Peronema canescens* have revealed a variety of secondary metabolites, which are responsible for its antioxidant, antibacterial, and anti-plasmodial effects [30,31]. Specifically, phytochemicals, such as β-sitosterol, phytol, β-amyrin, and several peronemins (A1, A3, B2, B3, C1, and D1) are integral to the plant’s biological activities and therapeutic potential, contributing to its anticancer, antibacterial, and antidiabetic properties (see Table 1 [1,2,3]).

Studies have revealed a variety of secondary metabolites, including alkaloids and flavonoids, which are responsible for its antioxidant, antibacterial, and antiplasmodial effects [30,31]. Also, additional compounds such as catechol, quinic acid, isovanillic acid, and guaiacol further enrich its phytochemical profile [27]. Other studies have reported the bioactivity of the ethanol extract of *Peronema canescens* leaves, which contain alkaloids, flavonoids, saponins, steroids, phenolics, and tannins [32,33]. This diverse array of phytochemicals underscores the therapeutic potential and traditional uses of *Peronema canescens*.

Pharmacological studies have demonstrated that *Peronema canescens* exhibits antioxidant, anticancer, anti-inflammatory, anti-hyperuric, and antidiabetic bioactivities [25]. First, the antioxidant bioactivity of *Peronema canescens* extract was demonstrated in a 70% ethanol extract using conventional methods and ultrasonic-assisted extraction (UAE), which has strong antioxidant activity with IC_50_ values of 50.78 and 53.50 μg/mL [34]. Next, a study reported that the secondary metabolites in the chloroform fraction of *Peronema canescens* leaves included alkaloids, terpenoids, steroids, flavonoids, and phenolics, which demonstrated cytotoxic activity (IC_50_) against HT-29 colon cancer cells ranging from 14.807 to 34.448 μg/mL. Chloroform subfraction 3 exhibited potential cytotoxic activity in human HT-29 cancer cells with an IC_50_ value of 14.807 μg/mL, offering promising prospects for further cytotoxicity studies [33].

Other in vivo and in vitro studies have also been conducted to assess macrophage phagocytic activity, leukocyte percentages, and total leukocyte counts in mice treated with *Peronema canescens* ethanol extract at doses of 200, 400, and 800 mg/kg BW, or 50 mg/kg for seven days. The in vitro study evaluated the viability and immunostimulatory activity of RAW 264.7 cells treated with *Peronema canescens* extract at 1, 10, and 100 μg/mL concentrations, confirming the safety and non-toxicity with cell viability above 90%. There were significant increases (*p* < 0.05) in macrophage activity, leukocyte counts, and cytokine levels (TNF-α and IL-6), confirming that *Peronema canescens* extract at the tested doses has notable immunostimulant effects both in vivo and in vitro [35]. Molecular docking studies revealed that peronemin, a key compound in *Peronema canescens*, potentially competed with IL-6 by inhibiting the TNF-α receptor [28]. Also, *Peronema canescens* exerted immunomodulatory bioactivity by increasing the phagocytosis index, total leukocytes, and percentage of leukocytes [36].

In a hyperuricemia model, the flavonoids in an ethanol extract of *Peronema canescens* inhibited the activity of xanthine oxidase (XO), reducing uric acid levels by up to 19% in mice [32], and demonstrated anti-inflammatory activity by reducing C-reactive protein levels [37,38]. *Peronema canescens* extract also reduced blood sugar levels, inhibiting α-glucokinase activity and increasing antioxidant activity, demonstrating its potential to help regenerate damaged pancreatic β cells [38,39].

The toxicity data regarding *Peronema canescens* extract are still limited. However, the ethanol extract of *Peronema canescens* is not toxic according to subchronic toxicity tests that were carried out for 28 days, which showed no signs of change or damage to haematological parameters, blood biochemistry, or the liver and kidney histopathology of the experimental animals [40].

**Table 1 biology-13-00744-t001:** Experimental research evaluating the active compounds/secondary metabolites and bioactivity of *Peronema canescens* Jack extract.

Research Aims and Design	Extraction Method	Solvent Used	Bioactive Compound	Biological Activity	Ref.
Identification of active compounds from PC leaves	Maceration	Acetone	β-sitosterol, phytol, β-amyrin, and peronemins (A2, A3, B1, B2, B3, and C1)	n/d	[1]
Extraction of active compounds in PC leaves	Maceration	Acetone and methanol	Caffeic acid and Peronemins (A2, A3, B1, B2, B3, C1, and D1)	n/d	[2]
In vitro antioxidant evaluation of PC metabolic extract	Maceration	Methanol	Alkaloid, flavonoid, saponin, steroid, and tannin	Antioxidant activity	[18]
Evaluation of anticancer activity of PC leaf extract on HT-29 cell line in vitro	Maceration and fractionation	Ethanol, hexane, chloroform, ethyl acetate, and methanol	Alkaloid, flavonoids, phenolic, steroids, and terpenoid	Best IC50 effect on HT-29 was 14.81 ug/mL in subfraction 3	[33]
Identification of the antioxidant activity of ethanol extract of PC in vitro	Maceration and UAE	Ethanol	Alkaloid, flavonoid, tannin, and saponin	Antioxidant activity	[34]
Identification of anti-inflammatory effects of PC leaf extract in vitro and in vivo	Maceration	Ethanol	n/d	Best immune stimulant effect was at a dose of 200 mg/kg BW mice	[35]
In vivo evaluation of the immunostimulant effect of PC leaf extract	Maceration	Methanol	n/d	Immunostimulant effect was observed starting from a dose of 25 mg/kg BW mice	[36]
Identification of anti-inflammatory effects of PC leaf extract in vivo	Maceration	Ethanol	n/d	Anti-inflammatory effects as evidenced by accelerated healing of the swelling caused by carrageenan	[37]
Evaluation of the anti-inflammatory effect of PC leaf extract in mice in vivo	Maceration	Ethanol	Alkaloid, flavonoid, phenolic, saponin, steroid, and tannin	Anti-inflammatory effects observed at a concentration of 15% in topical preparations	[38]
Evaluation of the antihyperuricemic effect of PC leaf extract in mice in vivo	Maceration and fractionation	Ethanol and n-hexane	Alkaloid, flavonoid, saponin, steroid, phenolic, tannin, and triterpenoid	Decrease in uric acid levels of 38.7% was obtained in EtOH extract at a dose of 500 mg/kg BW	[32]
In vivo antidiabetic evaluation of PC leaf extract	Maceration and fractionation	n-hexane, ethyl acetate, and methanol	Alkaloid, flavonoid, tannin, phenolic, saponin, and steroid	Methanol fraction reduced blood sugar levels in mice	[39]
Evaluation of compounds of PC leaves in vitro, and anti-inflammatory evaluation in vivo	Maceration and fractionation	Ethyl acetate, ethanol, and n-hexane	Alkaloid, flavonoid (apigenin), phenolic, and steroid (squalene)	Ethanol and n-hexane fraction demonstrated anti-inflammatory activity	[41]
Ethnopharmacology of the use of PC in the Dayak tribe in Indonesia	n/d	n/d	n/d	Young leaves of PC were used for the treatment of fever	[42]
Evaluation of the analgesic activity of ethanol extract of PC in vivo	Maceration	Ethanol	n/d	Analgesic effect observed at a dose of 600 mg/kg BW in mice	[43]

Abbreviation: HT-29: colon cancer cell line; n/d: not determined; UAE: ultrasonic-assisted extraction. PC: *Peronema canescens* Jack.

## 4. How Does an Immunomodulatory Agent Work?

An immunomodulatory agent is a compound or drug that targets the immune system, and its primary mechanism of action is to enhance or suppress the immune system to provide a beneficial therapeutic effect in treating autoimmune diseases, cancers, or infections. The mechanisms and sites of action of these drugs are varied according to the complexity of the immune system and the drug structure, as shown in Figure 1 [44].

Cytokines have an important role in communication between immune cells, and immunomodulatory drugs may target cytokines to inhibit or enhance their bioactivity [45,46]. TNF is a pro-inflammatory cytokine involved in various autoimmune diseases; TNF inhibitors include the immunomodulatory agents infliximab and adalimumab that bind to TNF-α to prevent it from interacting with its receptors (such as the TNF-α receptor), thereby reducing inflammation [47,48]. Interleukins are another class of cytokines with diverse biofunction. Several interleukin receptor inhibitors and antagonists, such as tocilizumab (an IL-6 receptor inhibitor) and anakinra (an IL-1 receptor antagonist), specifically target interleukin to modulate the immune response, particularly in conditions like rheumatoid arthritis and systemic juvenile idiopathic arthritis [49,50].

In addition to cytokines, there are immune checkpoints, which are regulatory pathways in the immune system that maintain self-tolerance and modulate the duration and amplitude of immune responses [51]. In cancer, tumour cells express immune checkpoint inhibitor ligands to evade the immune system’s ability to recognise and attack them [52]. Cytotoxic T-lymphocyte-associated protein 4 (CTLA-4) is an inhibitory receptor on T-cells. A tumour cell that expresses CD80/CD86 may interact with CTLA-4 and inactivate the immune cell. Ipilimumab, a CTLA-4 inhibitor, blocks this checkpoint, maintaining T-cell activation and proliferation, and enhancing the immune response against cancer cells [53]. In addition to CTLA-4, programmed death-1 (PD-1) is another inhibitory receptor on T-cells, while PD-L1 is its ligand, which is expressed on tumour cells. Drugs like pembrolizumab and nivolumab (PD-1 inhibitors), as well as atezolizumab (a PD-L1 inhibitor), block this interaction, restoring T-cell activity against tumours [54].

Immunosuppressive drugs are used to depress the immune response, particularly in autoimmune diseases and organ transplantation. Several examples of immunosuppressive drugs are cyclosporine and tacrolimus, which inhibit calcineurin [55], a key enzyme in T-cell activation, and glucocorticoids reduce inflammation [56]. Cyclosporin and tacrolimus prevent the transcription of IL-2 and other cytokines; these drugs reduce T-cell proliferation and activity [57]. Other drugs, like azathioprine and mycophenolate mofetil, inhibit purine synthesis, disrupting lymphocyte proliferation, which is critical for increasing the number of immune cells [58]. Sirolimus and everolimus inhibit the mammalian target rapamycin (mTOR), a kinase involved in T-cell proliferation and activation. These drugs reduce the immune response by inhibiting cell cycle progression and protein synthesis in T-cells [59].

Immunostimulants and immunoadjuvants enhance the immune response, making them useful in treating infections and certain cancers [60]. The development of immunostimulants is currently limited to cancer treatment due to the negative impact of increased immune cell activity. However, immunostimulant research is ongoing, evaluating indicators such as increased M1/M2 macrophage cells, neutrophils, natural killers, or lymphocytes [61].

**Figure 1 biology-13-00744-f001:**
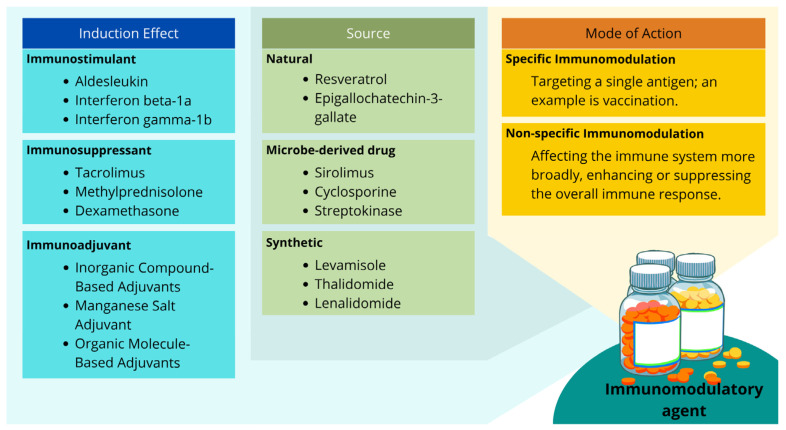
Classification of immunomodulatory agents based on their effects, mechanisms of action, and origin of development. Examples of generic and molecule names are obtained and compiled from various articles and databases (drug.com, Drugbank Online) [62,63,64].

## 5. A Potential Immunomodulatory Agent Based on a Secondary Metabolite of *Peronema canescens* Jack

To evaluate the potential of active compounds in *Peronema canescens* extract, it is necessary to understand how and where the extract works, especially in terms of its effect on the body’s immune system. This section discusses the bioactivity of *Peronema canescens* leaf extract and its potential as an immunomodulatory agent.

The exploration of immunomodulatory compounds is currently in the Asteraceae family [65]. From this family, *Echinacea purpurea* is the species best known to exert immunostimulatory effects due to its constituent alkaloids, polyphenols, and terpenoids [66]. Currently, there are eight plant metabolites with immunomodulatory effects in clinical trials or on the market including resveratrol, epigallocatechin-3-gallate, quercetin, colchicine, capsaicin, and andrographolide as immunosuppressants, and curcumin and genistein as immune stimulants [66,67].

In the leaf extract of *Peronema canescens*, alkaloids, flavonoids, and tannins were obtained from the maceration extraction method with solvents including ethanol, methanol, and ethyl acetate (Table 1). Meanwhile, phenolics and saponins were only obtained from the ethanol and methanol solvents, while steroids and triterpenoids were obtained from acetone and n-hexane fractions. Several compounds that were successfully isolated in several previous studies were the flavonoid apigenin, the steroid squalene, and the terpenoid peronemin [41]. Thus, the leaf extract of *Peronema canescens* has the potential to be a source of immunomodulatory agent compounds. However, due to the lack of information related to single compounds or other secondary metabolite isolates, further research is needed and there is the potential for the identification of novel compounds.

One of the secondary metabolites that has the potential to be a blueprint of *Peronema canescens* leaf extract is peronemin. This compound was first proposed by researchers from Fukuyama University—Isao Kitagawa and Partomuan Simanjuntak—in 1994 and 1996. Peronemin is a terpene with a diterpene structure and was identified in an acetone extract [1,2]. Peronemin C1 has since been shown to have a high affinity for binding to IL-6 and TNF-α receptors [28].

Based on the above discussion, *Peronema canescens* has extensive bioactivity. Immunomodulatory compounds are present in leaf extracts, although their specific structure has not been fully elucidated. However, these extracts contain immunostimulatory compounds that increase phagocytic activity and anti-inflammatory compounds, so further research is warranted to investigate possible synergistic effects or to isolate these compounds. However, the journey to develop this plant, maximise its pharmacological benefits, and integrate it into general healthcare might require several aspects and a long time. Plant pest management and cultivation methods are needed in terms of plant readiness, as well as toxicity tests and clinical trials. An understanding of commercial development and strategic recommendations for the community and government is mandatory in terms of prescription and precision medicine [68].

## 6. Future Perspectives

After discussing the potential of *Peronema canescens* (Sungkai) leaf extract, which has broad bioactivity and contains immunomodulatory agents, this section describes the future perspectives for the development of Sungkai leaf extract as a traditional or commercial medicine (Figure 2).

### 6.1. Plant Pest Management

Before discussing its extensive application as a raw material for pharmaceuticals, it is critical to understand how to effectively manage pests for the large-scale cultivation of Sungkai plants [69]. Fungal diseases, stem borers, and leaf-eating insects frequently pose a threat to Sungkai farming; therefore, integrated pest management (IPM) techniques must be applied including the use of natural pesticides made from plant extracts, biological control agents like predatory insects or parasitoids, and cultural measures like crop rotation and appropriate spacing. In addition to reducing damage and guaranteeing sustainable production, the routine monitoring and early detection of pest outbreaks might be helpful [70,71].

### 6.2. Cultivation Methods by Local Communities

Local communities in regions where *Peronema canescens* naturally thrives have developed traditional cultivation practises that can be leveraged and improved for broader agricultural use [72,73]. These practises typically involve the use of organic fertilisers [74], manual weeding, and natural pest control methods. Community-based cultivation also ensures the preservation of traditional knowledge and provides an economic incentive for local farmers [75,76]. Encouraging agroforestry practises to grow *Peronema canescens* alongside other crops can promote biodiversity and improve soil health, leading to more sustainable and resilient farming systems [77,78].

### 6.3. Potential for Toxicity Test and Clinical Trials

A safety evaluation of the potential immunomodulatory compounds from Sungkai extracts is necessary. Toxicity testing is an important stage to ensure that the therapeutic benefits of a preparation or drug are much greater than the damage caused. In addition, this test can also be used to evaluate possible damage such as carcinogenicity, immunotoxicity, genotoxicity, and teratogenicity. It can be conducted in vivo, in vitro, or in silico depending on the objectives and parameters being evaluated but still with logical, reliable scientific reasons and following the administration of the target country [79,80,81]. Specifically, toxicity tests can be categorised into primary pharmacology, secondary pharmacology, and chemically mediated toxicity to evaluate toxic doses, lethal doses, safety margins, reversibility, and toxic biomarkers, which are important to prevent adverse events in humans [82,83].

It was essential to conduct clinical trials in accordance with applicable administration and procedures to obtain full validation regarding the medicinal potential. Clinical trials are conducted after preclinical studies have shown promising results in terms of immunomodulatory, anticancer, and antidiabetic activities, and the effects and toxic doses are established [84,85,86,87]. There are three stages in clinical trials, as follows:Phase I Trials: Assessing the safety and tolerability of *Peronema canescens* extracts in a small group of healthy volunteers.Phase II Trials: Evaluating the efficacy and side effects in a larger group of patients with specific conditions such as autoimmune diseases, cancers, or diabetes.Phase III Trials: Confirming the efficacy, monitoring side effects, and comparing the extract with standard treatments in a larger population.Phase IV Trials: Drugs that have passed phase III testing are allowed to be distributed widely. Evaluation performed using a non-intervention model and research on side effects on consumers will be conducted.

Collaboration with research institutions and pharmaceutical companies can facilitate these trials, ensuring that the findings are robust and scientifically valid.

### 6.4. Commercial Development Potential

*Peronema canescens* has significant commercial potential due to its diverse therapeutic properties. The first stage of commercialisation is product development. Product development can be in the form of creating standardised extracts, capsules, and other formulas for medicinal use. In addition, its development into a food supplement can also be considered [88,89].

After obtaining a suitable product, the production process must be properly supervised following good manufacturing practise (GMP) guidelines, implementing stringent quality control measures to ensure the consistency, safety, and efficacy of the products [90,91]. Then, when the product is ready, marketing strategies can be developed to introduce *Peronema canescens* products to both local and international markets, including obtaining the necessary regulatory approvals and certifications. Regarding developing sales and the wider distribution of products, partnerships can be made with pharmaceutical companies, healthcare providers, and distributors [92].

### 6.5. Recommendations for Government Development

To fully harness the potential of *Peronema canescens*, the government should consider several strategic actions:Research and development: Educating the public about the advantages of *Peronema canescens* and its possible use in conventional medicine through awareness campaigns. Additionally, spending money on extensive research projects to investigate *Peronema canescens*’s complete spectrum of pharmacological advantages is also needed to provide more comprehensive information about the benefits and risks of this plant. Research related to this could include the exploration of phytochemicals, clinical trials, and the creation of standardised extracts for clinical use [93].Extension services: Along with its commercialisation, the government must also provide local farmers with information and instructions on the best ways to cultivate *Peronema canescens* based on GAP criteria [94], including post-harvest processing, sustainable farming methods, and pest control, to ensure economic equilibrium between supply and demand. The Indonesian Traditional Medicine Garden (TOGA) programme is implemented in Indonesia and could be used for Sungkai conservation [95,96].Financial Support: financial assistance (or low-interest loans) should be provided to farmers to cultivate *Peronema canescens* to help cover the initial costs of setting up plantations and purchasing the necessary equipment [97].Regulatory Framework: establish a regulatory framework to ensure the quality and safety of *Peronema canescens* products including setting standards for cultivation, harvesting, processing, and marketing [98].

## 7. Conclusions

*Peronema canescens* leaf extracts have various bioactivities including acting as immune modulators due to their constituent alkaloids, flavonoids, saponins, tannins, and terpenoids such as peronemin, apigenin, and squalane. There is potential for the development of these immunomodulatory agents for clinical use; however, the optimal utilisation of this plant requires collaboration between researchers, the government, industry, and the growers.

## Figures and Tables

**Figure 2 biology-13-00744-f002:**
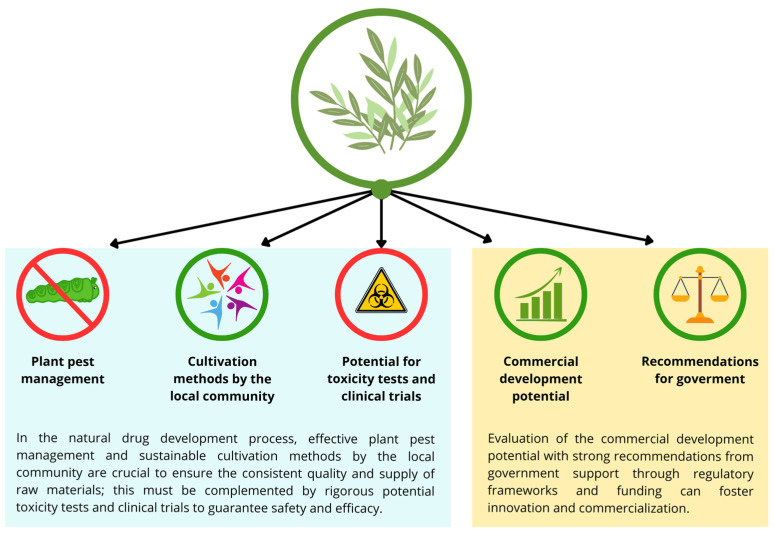
The importance of plant pest management, cultivation methods by the local community, potential toxicity tests and clinical trials, commercial development potential, and recommendations from government in drug development.

## Data Availability

All articles in Table 1 are free to access, but some of them are presented in Bahasa Indonesian. Additional explanations related to the bioactivity of immunomodulatory compounds in the extract can be accessed in related journals with individual subscriptions.

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
