# Peer review of "Peronema canescens as a Source of Immunomodulatory Agents: A New Opportunity and Perspective"

_biology, 2024, doi:10.3390/biology13090744_

Round 1
Reviewer 1 Report
Comments and Suggestions for Authors
This review provides a summary of recent findings on the biological activities and chemical composition of Peronema canescens. It also offers a more broad overview of immunomodulatory agents. The review begins with an introduction that effectively underscores the significance of compounds found in P. canescens, linking this to the growing interest in immunomodulatory agents, including those derived from this plant. The objectives of the review are clearly stated and logically structured.
Excluding the introduction and conclusion, the review is organized into seven distinct sections. These sections cover the botanical, phytochemical, and pharmacological aspects of P. canescens, as well as the mechanisms of action of immunomodulatory agents. The review also highlights the potential of phytochemicals as immunomodulatory agents.
However, I have several critiques and recommendations for improvement before publication:
1. Language and Terminology: The manuscript is in an early stage of development and contains multiple errors in grammar, syntax, and terminology. I strongly recommend a thorough review by a colleague whose first language is English.
2. Table 1: The current version of Table 1 is preliminary and needs restructuring. I suggest presenting the data in a more concise format with six columns: Extraction Method, Solvent Used, Identification Method, Bioactive Compounds, Biological Activity, and Reference.
3. Immunomodulatory Agents: The overview of the mechanisms of action would benefit from a visual representation. I recommend adding a chart that clearly explains these phenomena.
4. Section 5 (Lines 256-266): The paragraph that discusses a compound with potential immunomodulatory effects is the only one that adds value. The rest of the section repeats information about the chemical composition of different P. canescens extracts and should be revised.
5. Discussion (Section 6): The discussion is currently weak and lacks substantial value. It should either be deleted or integrated into the previous sections to strengthen the manuscript.
Additional Specific Recommendations:
- References should be added in the following locations: Lines 64, 134, 197, 237, 239, 250.
-The text in Lines 89-100, 155-157, 164-167, 169, 184-186 should be rewritten for clarity.
-Ensure that the point follows the reference citation, for example, "[24]." This applies to Lines 103, 105, 107, 114, 130, 137, 172, 182, 186, 194, 203, 212, 217, 243, 260, 266, 299, 303, 309, and others.
-The term "IC50" should be consistently used with proper units (Lines 137, 139).
-The terms "in vivo" and "in vitro" should be italicized (Lines 146, 147, 150, 155, 162).
-Use "p < 0.05" instead of "p<0.05" (Line 153).
-Correct "TNF-a" to "TNF-α" (Line 154), delete the word "about" (Line 158), and ensure "hematolo-gical" is correctly written as "hematolo-gical" (Line 171).
I hope these recommendations are valuable in enhancing the quality of the manuscript.
Comments on the Quality of English LanguageThe manuscript is in an early stage of development and contains multiple errors in grammar, syntax, and terminology. I strongly recommend a thorough review by a colleague whose first language is English.
Author Response
Please see the attachment. Thank you very much for your review report.

Reviewer 2 Report
Comments and Suggestions for Authors
A very interesting review that has collected all the information about the medicinal plant Peronema canescens, including the botanical profile and the study of the biological activity of the extracts. The authors also describe in detail the possible future prospects for using this useful plant, which is certainly of interest. I think this review is a good work and I am sure that it will be of interest to the readers of the journal Biology.
As minor comments, I can point out the following:
1) Line 137: The IC50 values are not indicated.
2) Line 139: The paragraph indicates cytotoxic activity only on one cancer cell line, but it would be interesting if the authors indicated what other cell lines were tested, even if they turned out to be inactive
3) Table 1. The first two lines contain n/a. Inactive for what? What type of activity was studied? Also in Table 1, n/a is found in columns 2 (Extraction method) and 3 (Active compounds). What does n/a mean in this case? Usually it means "not active". It would be better to replace to n/d (“not determined/not described”) in columns 2 and 3 and put the corresponding explanation of n/a and n/d at the end of the table.
4) all the Figures are of very poor quality (graphical abstract and Figure 1). The text is difficult to read. It is necessary to provide higher quality drawings.
Author Response
Please see the attachment. Thank your review report.

Round 2
Reviewer 1 Report
Comments and Suggestions for Authors The authors have thoroughly addressed the key concerns outlined in the previous review, resulting in substantial improvements to the manuscript. Consequently, I find the revised version to be suitable for publication in your journal.